# Markers of Cancer Cell Invasion: Are They Good Enough?

**DOI:** 10.3390/jcm8081092

**Published:** 2019-07-24

**Authors:** Tatiana S. Gerashchenko, Nikita M. Novikov, Nadezhda V. Krakhmal, Sofia Y. Zolotaryova, Marina V. Zavyalova, Nadezhda V. Cherdyntseva, Evgeny V. Denisov, Vladimir M. Perelmuter

**Affiliations:** 1Laboratory of Molecular Oncology and Immunology, Cancer Research Institute, Tomsk National Research Medical Center, 634009 Tomsk, Russia; 2Department of Cytology and Genetics, Tomsk State University, 634050 Tomsk, Russia; 3Department of Pathological Anatomy, Siberian State Medical University, 634050 Tomsk, Russia; 4Department of General and Molecular Pathology, Cancer Research Institute, Tomsk National Research Medical Center, 634009 Tomsk, Russia; 5Laboratory for Translational Cellular and Molecular Biomedicine, Tomsk State University, 634050 Tomsk, Russia; 6Department of Organic Chemistry, Tomsk State University, 634050 Tomsk, Russia

**Keywords:** cancer, invasion, invasive front, epithelial–mesenchymal transition, heterogeneity

## Abstract

Invasion, or directed migration of tumor cells into adjacent tissues, is one of the hallmarks of cancer and the first step towards metastasis. Penetrating to adjacent tissues, tumor cells form the so-called invasive front/edge. The cellular plasticity afforded by different kinds of phenotypic transitions (epithelial–mesenchymal, collective–amoeboid, mesenchymal–amoeboid, and *vice versa*) significantly contributes to the diversity of cancer cell invasion patterns and mechanisms. Nevertheless, despite the advances in the understanding of invasion, it is problematic to identify tumor cells with the motile phenotype in cancer tissue specimens due to the absence of reliable and acceptable molecular markers. In this review, we summarize the current information about molecules such as extracellular matrix components, factors of epithelial–mesenchymal transition, proteases, cell adhesion, and actin cytoskeleton proteins involved in cell migration and invasion that could be used as invasive markers and discuss their advantages and limitations. Based on the reviewed data, we conclude that future studies focused on the identification of specific invasive markers should use new models one of which may be the intratumor morphological heterogeneity in breast cancer reflecting different patterns of cancer cell invasion.

## 1. Introduction

Metastasis is a key feature of cancer and a “final chord” of the tumor progression [1]. The ability for metastasis enables tumor cells to leave the primary site and disseminate throughout the body, causing severe organ failure and leading to death. Understanding the mechanisms underlying metastasis is extremely important for the development of highly effective cancer therapies [2].

Metastasis is a complex process of stepwise events collectively termed the metastatic cascade and consisting of local invasion of tumor cells, intravasation to blood vessels, survival in the circulation, arrest at distant organs, extravasation into the parenchyma of distant tissues, micrometastasis formation, and metastatic colonization (macrometastasis) [1,2]. Invasion is the first step in the metastasis of tumor cells. From the morphological point of view, the invasion is a process during which malignant cells detach from the tumor mass, acquire the ability to actively move, and invade surrounding tissues through the adjacent basement membrane [3]. The interface of tumor and host tissue, in other words, the deepest rim of cancerous tissue grown in adjacent non-cancerous tissues, is called an “invasive front (edge)” [4]. Tumor cells constituting the invasive front are phenotypically different from cells in other tumor parts. Invasive front cells are believed to have a locomotor phenotype and demonstrate a variety of types and mechanisms of movement [5,6]. Tumor cells can move collectively or individually. The type of invasion depends on the molecular changes in tumor cells and the tumor microenvironment features [7,8,9,10]. The distinctive features of collective cell invasion include physical and functional relationships among tumor cells due to adhesion molecules as well as the presence of leader cells that are characterized by the mesenchymal phenotype and the ability to form lamellipodia, pull follower cells, and destroy the extracellular matrix (ECM) through production of proteases [11,12,13]. Interestingly, according some reports, invasive leaders do not express molecular features of epithelial–mesenchymal transition (EMT) [14], but exhibit a basal epithelial gene program, that is enriched in cytokeratin-14 and the transcription factor p63 [15,16].

Individual invasion can occur through mesenchymal and amoeboid cell migration mechanisms [17]. Sometimes, an intermediate amoeboid/mesenchymal (filopodial) cell migration mode is distinguished [18]. In mesenchymal movement, tumor cells exhibit a pronounced fibroblast-like phenotype, high expression of integrins, synthesis of proteolytic enzymes, and activity of small GTPases Rac1 and Cdc42 that are necessary to form lamellipodia and actomyosin contractions [7,12]. In amoeboid movement, cells are not capable of proteolysis and adhesion of the ECM but demonstrate the enhanced activity of the actomyosin machinery and the formation of cell membrane protrusions (blebs), which allow cells to squeeze through tight spaces in the surrounding matrix. Amoeboid movement directly depends on Rho/ROCK cell signaling and activity of type II myosin [13,17,19]. Tumor cells can transit from one cell migration phenotype to another via mesenchymal–amoeboid (MAT) and amoeboid–mesenchymal transition. The key role in these transitions is played by the balance of GTPases Rho and Rac, changes in expression of focal adhesion molecules and proteases, and ECM stiffness [13]. Importantly, the Rho/Rac feedback loop, particularly balanced relative high RhoA and Rac1, is also responsible for the hybrid amoeboid/mesenchymal phenotype in migrating cells [20].

EMT plays a key role in tumor dissemination. During EMT, tumor cells lose the epithelial phenotype and acquire the mesenchymal features and resistance to antitumor treatment; EMT also promotes immortalization and is involved in the prevention of apoptosis [21,22]. EMT is induced not only by molecular changes in tumor cells but also by cytokines and growth factors secreted by immune and stromal cells of the tumor microenvironment [23,24,25,26]. EMT may be incomplete (partial) when tumor cells still retain epithelial features but already acquire mesenchymal traits. During partial EMT, cells are described as a hybrid, with an intermediate epithelial/mesenchymal phenotype [27]. Partial EMT has been reported for both single tumor cells and tumor buds (groups of up to five cells) that are a variant of collective invasion [28]. The phenomenon “tumor budding” is regarded as a specific “signal” indicating the onset of cancer invasion and metastasis. The presence of tumor buds in the invasive front was found to be associated with increased metastasis and poor prognosis in various cancers [28,29,30,31,32,33,34].

Tumor cells can acquire the ability for migration not only through EMT but also through the so-called collective–amoeboid transition (CAT) when cells detach from the tumor mass and acquire an amoeboid phenotype rather than a mesenchymal phenotype. CAT is known to be regulated by the core regulatory circuits underlying EMT (miR-200/miR-34) and MAT (Rac1/RhoA) [35] and can be promoted by hypoxia-inducible factor 1 (HIF-1), which is accompanied by a decrease in E-cadherin expression [36]. However, CAT still remains a poorly understood phenomenon.

Active migration of tumor cells is not the only mechanism for invasive tumor growth. There is the so-called passive invasion when cells penetrate adjacent tissues under pressure from other tumor cells during proliferation (expansive growth) or due to an increase in the ECM density caused by the production of fibronectin and collagen by cancer-associated fibroblasts [37,38]. The fact that many circulating tumor cells are apoptotic [39,40], may be considered as indirect evidence of passive invasion, whereas active invasion is associated with viable cells [37].

Despite the fact that the mechanisms and types of cell migration and invasion have been described and studied quite well, there are currently no highly efficient and validated molecular markers for identification of migrating/invading tumor cells in tumors and, therefore, for assessment of their invasive potential. These markers could be used to identify patients at the high risk of distant metastasis and to prescribe therapy aimed at interrupting the metastatic process. In addition, these markers might represent targets for future therapeutics that block invasion and metastasis.

In this review, we systematized information about molecules that might be potential markers of tumor invasion and discussed the advantages and limitations of their use in clinical practice.

## 2. Potential Markers of Cancer Cell Invasion

The literature reports numerous studies describing various molecules that may act as markers of tumor cell invasion. Conventionally, they may be subdivided into several groups: ECM components, EMT, cell–cell and cell–ECM molecules, proteases, and actin cytoskeleton proteins (Table 1).

### 2.1. ECM Components

The first barrier to tumor cell invasion is the basement membrane that is a 100–300 nm thick ECM structure consisting of laminins, type IV collagen, and other non-cellular components, on which epithelial cells proliferate and differentiate [41,42,43,44]. Impaired integrity of the basement membrane is a histological marker indicating that carcinoma has acquired invasive properties [12,41,43]. A key component of the basement membrane, laminin-5, consists of α3, β3, and γ2 chains and plays a significant role in migration and invasion of tumor cells [43,45,46,47,48]. The interaction between laminin-5 and tumor cell integrins leads to the release of proteases and degradation of the basement membrane and ECM [43,47,49,50,51]. The laminin-5 γ2 chain monomer, which is considered as one of the most characteristic markers of invasion is found in the invasive front of different cancers [51,52]. For example, laminin γ2 expression combined with MMP-7 and EGFR expression in the invasive front is associated with gastric cancer aggressiveness [43]. In gastric cancer, cytoplasmic expression of laminin γ2 in tumor cells is related to lymph node metastasis and advanced stage [53]; in gallbladder cancer, stromal laminin γ2 expression is associated with a poor prognosis [54]. Laminin γ2 is also expressed in the invasive front of breast, pancreatic, colon, lung, and other cancers [46,51,52,55,56].

After penetrating the basement membrane, invading cells enter the ECM. Fibronectin is the major ECM component that plays a key role in the stimulation of cell growth, adhesion, and cell migration. On the one hand, fibronectin forms a physical barrier for migrating cells; on the other hand, its interaction with tumor cell integrins, mainly with α5β1, triggers ECM proteolysis through secreting MMP-2 and MMP-9 [42,112]. Fibronectin was demonstrated to be involved in the regulation of cell invasion and migration in various cancers [113] and expressed at the invasive front of oral and head and neck squamous cell carcinomas [60,61].

The tenascin C protein also belongs to ECM glycoproteins; however, it is mainly active during embryogenesis. In the adult body, tenascin C is found only in some types of connective tissue (tendons, ligaments, etc.). Interestingly, tenascin C is often expressed in the invasive front of breast, lung, liver, and gallbladder cancers, as well as melanoma, and is associated with a poor prognosis particularly decreased recurrence-free and overall survival and a high rate of metastasis [57,62].

Despite the proven association of basement membrane and ECM components with invasiveness, their role as markers of tumor invasion is ambiguous. For example, laminin γ2 expression is not always observed in the invasive front. According to Sentani [43], cytoplasmic laminin γ2 expression in the invasive front of gastric cancer occurs only in 25% of cases, and stromal expression is observed in 8% of cases. According to García-Solano [58], laminin γ2 expression in tumor buds at the invasive front of colorectal adenocarcinoma is found only in 17–57% of cases. In addition to the invasive front, laminin γ2 is also found in the basement membrane and cytoplasm of tumor cells, outside the invasive front [46]. Fibronectin and tenascin C are also expressed not only in the invasive front [57,59].

### 2.2. EMT Factors

EMT is common to almost all cancers, but the transition is rarely implemented in full [67]. Partial EMT is mainly typical of tumor cell clusters. However, there is evidence that single migrating cells may be in partial EMT. During partial EMT, tumor cells show co-expression of molecules of epithelial (E-cadherin, EpCAM, cytokeratin 7, miR-200, miR-34, etc.) and mesenchymal (N-cadherin, vimentin, ZEB, SNAIL, etc.) phenotypes. Cells in a partial EMT are capable of both adhesion and migration [27,28,67].

Overexpression of EMT markers is often observed in the invasive front of various cancers [63]. Nevertheless, molecules involved or associated with EMT are characterized by a low diagnostic value in assessing the invasive potential of tumors. Snail and Twist transcription factors are unstable molecules and undergo rapid proteasomal degradation [64,65]. In contrast, according to our data, Snail and Twist are totally expressed in breast tumor, without any selectivity in the invasive front [66]. Vimentin, which is considered a marker of the final EMT stage, may not be expressed in invasive carcinomas at all [67]. Furthermore, EMT is not always necessary for invasion and metastasis. In Snail and Twist knockout mice, tumor dissemination and the number of metastases are comparable to those in control mice [114]. Therefore, the presence of EMT cannot always answer the question whether the tumor cell migrates at a given time.

However, it should be understood that EMT is a complex process in which each step is thought to be regulated by a distinct set of transcription factors and molecular circuits overlapping to each other and generating specific phenotypes [115,116]. The picture is complicated by the fact that EMT transcription factors control other cellular events, including apoptosis and stemness [116]. Moreover, induction of an EMT transcription factor is known to be sufficient to induce single-cell dissemination without orchestrating the molecular EMT program and with retaining epithelial identity [16,117]. Thus, further studies are needed to explore molecular mechanisms underlying each EMT module, namely cell motility, and to find markers that could be used to assess the invasive potential of tumor cells. In addition, it is necessary to consider the fact that cells are capable of amoeboid and hybrid amoeboid/mesenchymal movement. Therefore, a perfect method for determining the invasive phenotype in tumor cells is the simultaneous assessment of markers of mesenchymal and amoeboid migration.

### 2.3. Cell–Cell and Cell–ECM Interaction Molecules

Adhesion molecules, such as integrins and the cadherin-catenin complex, are the key components of tumor invasion. Changes in the activity of cadherins, which are proteins involved in the formation of cell–cell contacts, is a characteristic feature of invasive growth. E-cadherin, which forms adherens junctions in an epithelial cell layer, is repressed by Snail, Slug, and Twist transcription factors during EMT [64]. The loss of E-cadherin and the nuclear localization of β-catenin, involved in signaling to the actin cytoskeleton [118], were observed in tumor cells at the invasive front in various cancers [69]. Nuclear accumulation of β-catenin in tumor cells in the invasive front and in vessels was found to be a powerful predictor of liver metastasis in colorectal cancer [70,71]. However, the loss of E-cadherin expression is probably not an indispensable prerequisite for invasiveness of tumor cells [72] and, therefore, cannot be used as a marker for invasive growth, at least for some cancers. Moreover, in some tumors, a loss of E-cadherin has been shown to be detrimental to invasion and metastasis. For example, the presence of E-cadherin is a specific feature of a highly aggressive form of breast cancer, inflammatory carcinoma, and needed for successful invasion and metastatic colonization of bone by tumor cells [119]. In this regard, analysis of more effective markers is needed to assess the invasive tumor potential, along with markers of amoeboid movement, as mentioned above.

The key event initiating production of metalloproteinases is the interaction of integrins with ECM components. The main ligands for integrins are fibronectin (α5β1, αvβ3, and α4β1 integrins), collagens (α1β1, α2β1, and α11β1), and laminins (α2β1, α3β1, α6β1, and α6β4) [41,64,120,121,122,123]. For example, α3β1 integrin activates MMP-9 synthesis through interaction with laminins and triggers reorganization of the actin cytoskeleton [124]; α6β1 is involved in tumor invasion via activation of the urokinase plasminogen activator (uPA) receptor and MMP-2 [125]. Laminin-5 is the best-characterized ligand for α3β1 integrin. α6β4 integrin is involved in the regulation of tumor cell migration through activation of the Rho-A signaling cascade [121]. Binding of fibronectin to α5β1 integrin activates MMP-1 and stimulates migration through the ILK/Akt and GSK3β/Snail/E-cadherin signaling pathways [121,126]. Fibronectin-mediated migration is also associated with αvβ3 integrin. αvβ3 integrin is involved in activation of MMP-2 [127] and, under stress conditions, can trigger a ligand-independent signaling cascade leading to activation of NF-κB and Slug, acquisition of a stem phenotype, and promotion of migration [126].

Expression of integrins changes during tumor progression and is often elevated in the invasive front of tumors: αvβ3 in melanoma [75], αvβ6 in colon and head and neck cancers [73,76], and α6β4 in non-small cell lung cancer [74]. Furthermore, high expression of integrins in tumor cells may promote metastasis. For example, α2β1 enhances metastasis of rhabdomyosarcoma in nude mice after intravenous or subcutaneous injection [128], whereas α3β1 promotes lung metastasis through binding to laminin-5 in an exposed basement membrane in the pulmonary vasculature [50].

Signaling pathways activated by different integrins may lead to the same biological effects, while an individual contribution of each of the integrins is different. In neuroblastoma, tumor cell migration can be activated either via FAK-mediated α5β1 integrin signaling or via a FAK-independent pathway involving α4β1 integrin. Both signaling pathways lead to the induction of Src family protein kinases [129,130].

The use of integrins as markers of invasive growth is complicated by the fact that the same integrins can participate in both invasion and other biological processes [78]. For example, α6β1 integrin, apart from involvement in tumor invasion, also participates in Ca^2+^ signaling [131] and platelet adhesion upon damage to the vascular wall [132].

There is evidence that changes in expression of other cell interaction proteins may be a marker of invasive tumor cells. Galectins, membrane glycoproteins, bound to integrins, laminins, and fibronectin, are used by cells to interact with each other and with the ECM [47,133]. Galectin-1 is involved in the regulation of cell adhesion and migration, on the one hand, through stimulation of MMP-2 and MMP-9 and, on the other hand, through activation of a small Rho GTPase Cdc42, which promotes the formation of actin filopodia. Increased expression of galectin-1 is associated with high invasiveness of lung adenocarcinoma and observed in the invasive front of oral squamous cell carcinoma and glioblastoma [47,79,80]. However, galectins have effects not only on tumor cells but also on immune cells promoting inflammation or dampening T cell-mediated immune responses [77]. The L1 cell adhesion molecule (L1CAM), which is involved in β-catenin/TCF signaling, is necessary for cell migration and invasion. Normally, L1CAM is present only in the nervous tissue, but its expression is induced in tumor cells. Increased expression of L1CAM was found in many cancers, including the invasive front of colorectal and pancreatic cancers [81,82]. Nevertheless, L1CAM can have a static function as a cell adhesion molecule and its expression is associated with good cancer prognosis [83,134].

### 2.4. Serine Proteases and Matrix Metalloproteinases

One of the main systems responsible for ECM proteolysis is the plasminogen activation system that triggers a powerful serine protease, plasmin. The central component of this system is the uPA and its receptor (uPAR), the interaction of which stimulates proteolysis of plasminogen to plasmin [135,136]. uPA is believed to play a significant role in tumor invasion and metastasis [135,136,137]. Experiments in model animals demonstrated that inhibition of uPA and/or the uPA/uPAR interaction slows down metastasis [135]. In contrast, expression of uPAR is associated with tumor invasion and is found in stromal and tumor cells in the invasive front of oral and skin squamous cell carcinomas [84,85].

Metalloproteinases are involved in proteolytic degradation of the basement membrane and ECM. MMP-7 activates MMP-2 and MMP-9 gelatinases exhibiting proteolytic activity against collagen IV, laminins, proteoglycans, and fibronectin [138]. Expression of MMPs is observed during cancer cell invasion [13,41]. MMP-7-positive tumor cells are predominantly found in the invasive front of gastric cancer, while their number is much higher in aggressive and late-stage tumors [90,91]. MMP-7 is also expressed in the invasive front of colon cancer and correlates with tumor stage [56,91,92]. Elevated MMP-2 and MMP-9 levels are observed in the invasive front of melanoma, endometrial cancer, and ovarian cancer [89,93]. High MMP-2 and MMP-9 expression is also observed in the invasive front of head and neck squamous cell carcinoma [88,94]. Assessment of MMP-2 and MMP-9 expression in the invasive tumor front may be helpful in the differentiation of verrucous carcinoma and squamous cell carcinoma of the oral cavity [139].

However, increased expression of uPA and MMPs is not a unique feature of invasive tumor cells and may be observed in other physiological processes. The components of the uPA system can be involved in the early stages of tumor formation and can increase cell proliferation, inhibit apoptosis, etc. [86]. MMPs are mediators between tumor cells and the microenvironment [87]. MMP-9 produced by inflammatory cells is involved in the proteolytic activation of anti-inflammatory cytokines TGF-β2 and TGF-β3, and MMP-2 and MMP-14 participate in the activation of TGF-β1 [87,140,141]. MMP-2, MMP-9, and MMP-14 indirectly modulate TGF-β activity by cleaving an ECM component, the latent TGF-β binding protein 1 [87,142]. MMP-7 inhibits apoptosis and reduces the efficacy of chemotherapy by cleaving Fas ligands on the surface of cells exposed to doxorubicin [87,143]. MMP-2 and MMP-9 are also involved in the regulation of angiogenesis and lymphangiogenesis [87]. MMP-9 secreted by inflammatory cells modulates bioavailability of VEGF to the VEGFR2 receptor [87,144]. Experiments in mice demonstrated the role of MMP-9 in triggering the angiogenic switch and in vasculogenesis [87,145,146]. Therefore, the multifunctionality of MMPs reduces their significance as markers of invasive growth.

### 2.5. Actin Cytoskeleton Proteins

Proteins involved in actin cytoskeleton remodeling play an important role in the mechanisms of tumor cell migration and invasion [147]. The ezrin protein is a connecting link between actin filaments and membrane proteins involved in cell–cell adhesion and migration [148]. Ezrin was demonstrated to be localized together with the podoplanin in filopodia, stimulating cellular invasion [149], and expressed in the invasive front of lung cancer [95]. Many studies reported that upregulation of Ezrin is a negative prognostic factor in various cancers. However, there is an opposite data indicating the involvement of negative or reduced expression of Ezrin in cancer progression [96]. This contradiction can be explained by the fact that Ezrin is implicated in the regulation not only of cell motility but also of cell adhesion, ion channels, cell proliferation, etc. [150].

The WAVE2 protein is involved in actin filament reorganization and lamellipodia formation and was shown to colocalize with Arp2 at the invasive front of breast cancer [97,147].

Cortactin regulates cortical actin cytoskeleton dynamics by stabilizing F-actin networks and promoting actin polymerization via activating the Arp2/3 complex [47,151]. According to in vitro and in vivo experiments, cortactin promotes invasion of head and neck tumors [151], and its high expression is found in the invasive front of oral and laryngeal tumors [98,99].

The MENA protein regulates actin polymerization and cell migration. An elevated level of the MENA^inv^ isoform, which is involved in the formation of invadopodia due to phosphorylation of cortactin and activation of the N-WASP/Arp2/3 complex, is found in invasive cells of human tumors and animal tumor models and is associated with a high risk of metastasis [100,152,153].

Fascin-1 is an actin-binding protein involved in filopodia formation. It is highly expressed in nervous tissue and is normally absent in epithelial cells. However, a high level of fascin-1 is found in many malignant neoplasms of the liver, gallbladder, stomach, intestines, lung, breast, etc., and is a marker of poor prognosis [154,155]. Increased expression of fascin-1 is found in the invasive front of liver, colon, cervical, and endometrial cancers and is associated with a high risk of metastasis [101,102,103,104].

### 2.6. Other Proteins

In the invasive front, there are highly proliferating tumor cells, which probably facilitate the more efficient dissemination of the tumor. Expression of Ki-67, a cell proliferation marker, was shown to be elevated in the invasive front of oral and endometrial cancers [6,105,106]. In breast cancer, nuclear expression of Ki-67 is two-fold higher in the invasive front than in other parts of the tumor and is associated with metastasis to bones and liver [107]. Increased proliferation of tumor cells in the invasive front is also indicated by elevated expression of FGFR2 that is involved in the induction of signaling pathways affecting division, growth, and differentiation of cells, as demonstrated in colorectal and cervical cancers [109,110]. However, there are also contradictory data on negative expression Ki-67 or the absence of differences in its level between the invasive front and the tumor center in oral and colorectal cancers [56,69,108]. Moreover, FGFR2 is a multifunctional protein that regulates different biological processes such as proliferation, differentiation, etc. [111].

At first glance, the prevalence of cell proliferation in the invasive front is in contradiction to the data that invading tumor cells are enriched in EMT markers [63] because EMT typically associates with cell cycle arrest [156]. However, in the invasive front, EMT-cell cycle connection can be broken. In other words, instead of “go-or-grow”, tumor cells follow “go-and-grow” behavior [115,157].

The search for tumor invasion markers is an important issue aimed at assessing the risk of cancer metastasis. The role of the discussed molecules as invasive markers is controversial in most cases. Most of these molecules are involved not only in invasive growth but also in processes not related to cell migration. Nevertheless, some molecules such as WAVE2, cortactin, MENAinv, and fascin-1 are promising candidates for future studies of their roles as cancer cell invasion markers. In any case, the search for more specific markers of invasive growth is needed. In this regard, we think that the emphasis on intratumor morphological heterogeneity typical of many cancers may be very productive. In particular, investigation of the molecular make-up of various invasive tumor structures may enable identification of new molecules associated with invasion of tumor cells.

## 3. Intratumor Morphological Heterogeneity as a Model for Studying Cancer Cell Invasion

Based on more than 10-year morphological studies and detailed analysis of various structural features of invasive carcinoma of no special type of the breast (IC NST, previously classified as invasive ductal carcinoma), we have concluded that there are two types of tumors: Nonstructural and structural (Figure 1). Nonstructural breast carcinomas are characterized by a monomorphic pattern and are represented by large solid areas connected to each other, with thin layers of stromal elements (Figure 1).

Structural tumors are characterized by a polymorphic pattern and a pronounced phenotypic variety of the infiltrative (invasive) and stromal components (Figure 1). In other words, structural tumors demonstrate significant morphological heterogeneity. In initial attempts to determine the potential morphological IC NST features associated with cancer progression, we identified five main types of the invasive component in the tumor: Tubular, alveolar, solid, and trabecular structures, and discrete groups of tumor cells [158,159,160,161]. The tubular structures are tube-shaped and lumen-containing arrangements of single rows of rather monomorphic tumor cells with round monomorphic nuclei. The alveolar structures are clusters of round or slightly irregular tumor cells of different sizes, often with polymorphic nuclei. The number of cells in alveolar structures varies from 5–20. The solid structures are represented by large masses differing in size and shape, which consist of either small tumor cells with moderate cytoplasm and monomorphic nuclei or large cells with abundant cytoplasm and polymorphic nuclei. Although solid groups of tumor cells are a characteristic feature of nonstructural breast tumors, they are also observed in structural carcinomas. The trabecular structures are represented by either a single row of tumor cells (≥5 cells) or arrangements consisting of two rows of closely related monomorphic cells with moderate cytoplasm, which are parallel to each other. The discrete groups consist of small cell clusters (up to five cells) and single tumor cells (Figure 1). The size and shape of these cells and nuclei vary significantly [158,159,160,161].

Different morphological structures were shown to represent transcriptionally distinct tumor cell populations differing in the number of CD44^+^CD24^−^ cancer stem cells, epithelial and mesenchymal features, and enrichment of cancer invasion signaling pathways [160]. Tubular and alveolar structures are similar in gene expression and demonstrate co-expression of epithelial and mesenchymal markers. The solid structures retain the epithelial features but demonstrate an increase in the mesenchymal traits and collective cell migration hallmarks. Trabecular and discrete groups are enriched in mesenchymal genes and cancer invasion pathways. CD44^+^CD24^−^ cells are less common in the discrete groups and more abundant in the alveolar and solid structures [160]. Taken together, these data suggest that different morphological structures demonstrate varying degrees of EMT: From low in tubular, alveolar, and solid structures to advanced in trabecular and discrete groups of tumor cells [160].

The intratumor morphological heterogeneity of breast cancer is not an occasional phenomenon and is strongly associated with disease prognosis and therapy efficacy. Breast tumors with either alveolar or trabecular structures are characterized by a high rate of lymph node metastasis [161,162]. In neoadjuvant chemotherapy (NAC), tumors with alveolar or trabecular structures often demonstrate a poor response [162,163] and an increased risk of distant metastasis [162]. NAC-treated patients with alveolar or trabecular structures in breast tumors have decreased metastasis-free survival [162].

In a longitudinal study of the morphological, molecular genetic and clinical features of breast cancer, we have clearly seen that the differences are present not only in the structural pattern of tumor tissue. It has become obvious that breast carcinoma is characterized by pronounced intratumor morphological heterogeneity when morphologically similar and almost identical structures can exhibit completely different expression profiles, and it may not be ruled out that this phenomenon may somehow affect the behavior of tumor [66]. This conclusion prompted us to differentiate in more detail the previously described morphological structures.

A morphological analysis of structural IC NSTs revealed significant diversity and variability in solid groups of tumor cells, among which we identified two different variants: Solid structures with large torpedo-like sprouts and solid structures with small bud-like sprouts (Figure 1). The first variant is represented by various differently-sized, merging solid areas of tightly packed tumor cells connected with each other. In these structures, there are elongated, mostly triangular sprouts consisting of two–three parallel cell rows. The base of torpedo-like sprouts is always pointed out to the body of solid structures, while the tip, consisting of one–three tumor cells, penetrates to different depths to the stroma. Importantly, torpedo-like sprouts can be presented as structures independent of solid groups of tumor cells (Figure 1). Another variant of solid structures is represented by the large masses of tumor cells. However, a distinctive feature is that any edge of a solid structure comprises rounded or spherical bud-like sprouts consisting of five–seven atypical cells penetrating to the stroma (Figure 1).

Thus, the structural diversity of the infiltrative component and the pronounced intratumor morphological heterogeneity in IC NST represent an attractive model for investigation of tumor cell invasion. The solid structures both with large torpedo-like and small bud-like sprouts, as well as trabecular structures, may be considered as a morphological manifestation of collective cell invasion. Discrete groups of tumor cells, mainly single tumor cells, are an example of individual cell invasion.

## 4. Conclusions

Invasion is a key event towards the acquisition of the metastatic phenotype by tumor cells and an attractive target for anticancer therapy aimed at the prevention of metastasis. In in vitro studies, EMT has been proved to play an important role in the appearance of migrating and invading tumor cells. However, the cell movement mechanisms working in vitro are frequently not related to the invasive growth in vivo. Molecules that have been identified in vitro to be involved in cancer cell invasion do not demonstrate selective expression at the invasive front or at the tips of invasive structures where tumor cells are rather motile. Moreover, the expression of these molecules does not often demonstrate clinical significance for the prediction of cancer metastasis risk. Thus, the question how to identify invading tumor cells in human cancer specimens remains unanswered. In this regard, new effective models should be developed to investigate the mechanisms of cancer cell invasion. In our opinion, one of these models, at least in case of breast cancer, can be intratumor morphological heterogeneity which is a manifestation of different patterns of tumor cell invasion. The investigation of the molecular make-up of invasive structures of tumor cells and their microenvironment may provide valuable information about new molecules involved in the invasive growth and may identify novel prognostic markers and therapeutic targets.

## Figures and Tables

**Figure 1 jcm-08-01092-f001:**
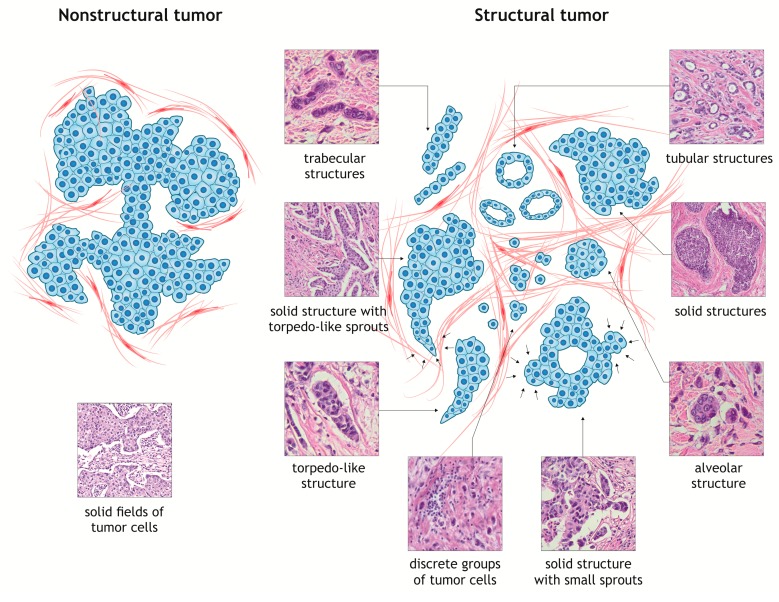
Two types of breast carcinomas based on a structural pattern. Nonstructural breast carcinomas are represented by large solid fields of cells connected to each other. Structural breast carcinomas are characterized by a phenotypic variety of the infiltrative (invasive) component, represented by certain types of morphological arrangements of tumor cells: Tubular structures, solid structures with small sprouts, solid structures with large torpedo-like sprouts, alveolar structures, torpedo-like structures, trabecular structures, and discrete groups of tumor cells. The images of hematoxylin and eosin-stained sections were obtained from the database of the Department of Pathological Anatomy, Siberian State Medical University, Tomsk, Russia.

**Table 1 jcm-08-01092-t001:** Potential markers of cancer cell invasion.

Markers		Functions	Expression at the Invasive Front	Limitations
ECM components	Laminin-5, γ2 chain	ECM components, triggering MMP production through interaction with integrins	Breast, pancreatic, colon, lung, and other cancers [46,51,52,55,56]	Expression not only in the invasive front, but in other regions of the tumor [43,46,57,58,59]
	Fibronectin		Oral and head and neck cancers [60,61]	
	Tenascin C	Modulation of cell adhesion	Melanoma, breast, lung, liver, and gallbladder cancers [57,62]	
EMT molecules	Snail, Twist, vimentin	EMT induction and regulation	Various cancers [63]	Snail and Twist: Unstable molecules [64,65], total expression in breast tumors [66]. Vimentin may not be expressed in invasive carcinomas [67]
Cell–cell and cell–ECM interaction molecules	Cadherin-catenin complex	Adherens junctions	Colorectal, oral, and basaloid carcinomas (loss of E-cadherin and nuclear localization of β-catenin) [68,69,70,71]	In some tumors, loss of E-cadherin is not indispensable for invasive growth [72]
	Integrins	Cell–ECM adhesion, involvement in MMP production	Melanoma (αvβ3), colon (αvβ6), head and neck (αvβ6), and lung (α6β4) cancers [73,74,75,76]	Involvement in other biological processes [77,78]
	Galectin 1	Modulation of cell–cell and cell–ECM interactions	Oral and lung cancers, glioblastoma [47,79,80]	
	L1CAM	Cell adhesion	Colorectal and pancreatic cancers [81,82]	Dualistic role in cancer progression [83]
Serine proteases and MMPs	uPA	Proteolysis of plasminogen to plasmin	Oral and skin carcinomas [84,85]	Involvement in other biological processes [86,87]
	MMPs	ECM proteolysis	Melanoma (MMP-2), colorectal (MMP-7), gastric (MMP-7), endometrial (MMP-2, 9), ovarian (MMP-2, 9), and head and neck (MMP-2, 9) cancers [56,88,89,90,91,92,93,94]	
Actin cytoskeleton proteins	Ezrin	Actin polymerization, cytoskeletal dynamics	Lung cancer [95,96]	Involvement in other biological processes. Contradictory data on the role in cancer progression [96]
	WAVE2		Breast cancer [97]	-
	Cortactin		Oral and laryngeal cancers [98,99]	-
	MENAinv		Breast cancer [100]	-
	Fascin-1		Liver, colon, cervical, and endometrial cancers [101,102,103,104]	-
Other proteins	Ki-67	Cell proliferation	Breast, oral, and endometrial cancers [6,105,106,107]	Contradictory data on the level of Ki-67 expression at the invasive front [56,69,108]
	FGFR2	Cell division, growth and differentiation	Colorectal and cervical cancers [109,110]	Involvement in other biological processes [111]

ECM, extracellular matrix; EMT, epithelial–mesenchymal transition; MMPs, matrix metalloproteinases.

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
