# Peer review of "Markers of Cancer Cell Invasion: Are They Good Enough?"

_jcm, 2019, doi:10.3390/jcm8081092_

Round 1
Reviewer 1 Report
The authors exhaustively covered all the mechanism currently known to be involved in breast cancer metastasis. My only suggestion is to add a table/figure that summarized all the described mechanisms.
Author Response
We thank the Reviewer for the work on the manuscript and the useful suggestion.
Reviewer:
The authors exhaustively covered all the mechanism currently known to be involved in breast cancer metastasis. My only suggestion is to add a table/figure that summarized all the described mechanisms.
We added the table that summarizes cancer invasion markers described in the manuscript, their functional role, and expression at the invasive front.
Reviewer 2 Report
The article “Markers of Cancer Cell Invasion: Are They Good 2 Enough” by Tatiana S. Gerashchenko et al ., is a well written and interesting article especially in the context of cancer cells metastasis as invasion is a major component of hallmarks of cancer and preventing invasion is a major hurdle in the current design of therapies against cancer. The authors describe in detail the ECM components, epithelial to mesenchymal transcription factors and proteins involved in adhesion and their contribution in invasion and metastasis. Moreover, the authors also describe the intratumor heterogeneity in understanding cancer cell invasion as this is an evolving field and it could be of a great interest to the readers of this journal. Although the article is well written, I have few minor additions to be considered by the authors to improve the overall quality of the article.
Minor Comments.
1.Although the invasion is a major and important part of the metastatic cascade and papers topic, the authors should mention all the steps in cancer progression such as proliferation, invasion, extravasation, intravasation and metastasis for the readers in the introduction for the general understanding for the readers on the steps in cancer progression and metastasis.
2.Cite the images of the H and E staining from where they were obtained in the figure legend.
3.Could you please elaborate on the concept of use of these markers as biomarkers to stratify patients in the clinic and provide the respective treatment.
Author Response
We thank the Reviewer for finding the manuscript well written and interesting and the constructive comments.
Reviewer:
The article “Markers of Cancer Cell Invasion: Are They Good 2 Enough” by Tatiana S. Gerashchenko et al ., is a well written and interesting article especially in the context of cancer cells metastasis as invasion is a major component of hallmarks of cancer and preventing invasion is a major hurdle in the current design of therapies against cancer. The authors describe in detail the ECM components, epithelial to mesenchymal transcription factors and proteins involved in adhesion and their contribution in invasion and metastasis. Moreover, the authors also describe the intratumor heterogeneity in understanding cancer cell invasion as this is an evolving field and it could be of a great interest to the readers of this journal. Although the article is well written, I have few minor additions to be considered by the authors to improve the overall quality of the article.
Minor Comments.
1. Although the invasion is a major and important part of the metastatic cascade and papers topic, the authors should mention all the steps in cancer progression such as proliferation, invasion, extravasation, intravasation and metastasis for the readers in the introduction for the general understanding for the readers on the steps in cancer progression and metastasis.
The information regarding the metastatic cascade and the corresponding references have been added in the introduction of the manuscript (page 1, lines 38-41).
2. Cite the images of the H and E staining from where they were obtained in the figure legend.
The images of H&E stained sections were obtained from the database of Department of Pathological Anatomy, Siberian State Medical University, Tomsk, Russia. The corresponding information has been added in the figure legend (page 8, lines 305-307).
3. Could you please elaborate on the concept of use of these markers as biomarkers to stratify patients in the clinic and provide the respective treatment.
We added the information regarding the potential use of markers of cancer cell invasion in clinical practice (page 2, lines 93-95).